# Transparent Heat Shielding Properties of Core-Shell Structured Nanocrystalline Cs_x_WO_3_@TiO_2_

**DOI:** 10.3390/nano12162806

**Published:** 2022-08-16

**Authors:** Luomeng Chao, Changwei Sun, Jiaxin Li, Miao Sun, Jia Liu, Yonghong Ma

**Affiliations:** College of Science, Inner Mongolia University of Science and Technology, Baotou 014010, China

**Keywords:** tungsten bronzes, nanocrystals, core-shell structure, heat-shielding materials

## Abstract

Nanocrystalline tungsten bronze is an excellent near-infrared absorbing material, which has a good potential application in the field of transparent heat shielding materials on windows of automobiles or buildings, but it exhibits serious instability in the actual environment, which hinders its further application. In this paper, we coated the Cs_x_WO_3_ nanoparticles with TiO_2_ to prepare core-shell structured Cs_x_WO_3_@TiO_2_, and its crystal structure and optical properties were studied. The results show that the surface of Cs_x_WO_3_ nanoparticles is coated with a layer of TiO_2_ particles with the size of several nanometers, and the shell thickness can be adjusted by the amount of Ti source. The measurement of optical properties illustrates that TiO_2_-coated Cs_x_WO_3_ exhibits good stability in actual environment, and its transparent heat shielding performance will decrease with the increase in TiO_2_ shell thickness. This work provides a new route to promote the applications of tungsten bronze as heat shielding materials.

## 1. Introduction

Nearly half of the solar radiation energy comes from near infrared light (NIR) in the range of about 760–2500 nm. Therefore, if the glass of buildings or automobiles can block NIR light while maintaining high transmittance of visible light, it can effectively reduce the room temperature, thus reducing the utilization of air conditioning and achieving the purpose of energy saving and emission reduction. At present, the most common commercial transparent heat shielding glass is Low-E glass or ITO glass [1,2], but the popularity of these materials is not very high because of its complex preparation process and high cost. Therefore, the current research has focused on new transparent heat shielding materials such as rare-earth hexaborides, VO_2_ or tungsten bronzes [3,4,5,6,7,8,9,10,11]. All of these materials have good application prospects in the field of transparent heat shielding materials, but there are also some problems. It is difficult to prepare nanocrystalline rare-earth hexaborides and the cost is high; high phase transition temperature of 68 °C and low visible transmittance hinders the further development of VO_2_; although tungsten bronze has excellent properties, its stability in the practical environment is insufficient.

The chemical formula of tungsten bronzes can be written as M_x_WO_3_, where the M represents alkaline earth metal, alkali metal, ammonium or rare earth metal ion. When the M cations are inserted into the whole gap of M_x_WO_3_, then x = 1. In the actual environment, M_x_WO_3_ is easily oxidized, and M^+^ escapes from the particle and forms WO_3_ on the surface, which leads to serious instability of NIR absorption of M_x_WO_3_ [12]. A good way to solve this problem is coating M_x_WO_3_ nanoparticles with suitable materials. Jin et al. prepared core-shell structured Cs_x_WO_3_@SiO_2_ and Cs_x_WO_3_@ZnO nanoparticles and achieved high stability of NIR absorption [13,14]. In the previous theoretical calculation, we found that TiO_2_-coated Cs_x_WO_3_ (CWO) also has good transparent heat shielding properties [15]. Considering the better stability of TiO_2_, we synthesized core-shell structured nanocrystalline CWO@TiO_2_ in this work, and the stability and optical behavior of nanoparticles was investigated in detail.

## 2. Materials and methods

### 2.1. Preparation of CWO Nanoparticles

A total of 0.7554 g of cesium hydroxide monohydrate (CsOH·H_2_O) was added to 300 mL benzyl alcohol (C_7_H_8_O) solution and stirred for half an hour. Then, an appropriate amount of tungsten hexachloride (WCl_6_) was added and the WCl_6_ concentration of precursor solution was kept at 0.015 M. After that, the orange precursor solution was heated in an autoclave at 200 °C for 4 h. Finally, the obtained blue powder was washed with alcohol and deionized water several times, and dried in a vacuum at 60 °C for 2 h.

### 2.2. Preparation of CWO@TiO_2_ Nanoparticles

An amount of 3 g CWO powder was added to 400 mL ethanol and dispersed by sonication for 30 min. Then, a different amount (0.2, 0.4 and 0.6 mL) of titanium isopropoxide (TTIP) was added to the solution for forming TiO_2_ shell on the surface of CWO particles. After that, deionized water was added to the solution with strong stirring. Then, the mixture was heated in the autoclave at 200 °C for 18 h. Finally, the product was washed several times with deionized water, and dried at 50 °C in vacuum overnight. According to the different amount of TTIP used in the reaction process, the obtained three samples are expressed as CWO@0.2TiO_2_, CWO@0.4TiO_2_ and CWO@0.6TiO_2_, respectively.

### 2.3. Fabrication Process for CWO@TiO_2_ Coated Glass

The heat-shielding glass was prepared by the spin coating method. First, 0.2 g CWO@TiO_2_ was dispersed in 20 mL ethanol solution by ultrasonic for 30 min, then 5 g polyvinyl butyral (PVB) resin was added under strong stirring for 20 min to obtain coating slurry. After spin-coating with a centrifugal speed of 2000 rad/min for 40 s, the coated glass was kept at 40 °C for 1 h to remove residual liquid.

### 2.4. Discrete Dipole Approximation Method

Discrete dipole approximation method (DDA) is an effective way to calculate the far-field and near-field optical responses of nanoparticles with complex refractive indexes and arbitrary geometries [16,17,18,19]. In this work, an open-source Fortran-90 code DDSCAT 7.3 applying the DDA method [20] was used to simulate the extinction efficiencies of CWO@TiO_2._ The program has given ellipsoid, regular tetrahedron, cuboid, cylinder, hexagonal prism, regular tetrahedron and other structural models. These models can be combined with the dielectric function of the corresponding material to calculate the extinction, absorption and scattering efficiency. Specific parameters such as the effective radius of particles, the number of dipoles and the wavelength range should be set in the program during calculation. In our calculation, a complex dielectric constant of CWO measured by Sato et al. was used to simulate the optical response [21] and the dielectric constants of TiO_2_ were obtained from the diel files of DDSCAT 7.3. The calculated wavelength range was 300 nm–2500 nm with 100 steps; The effective radius of the CWO@TiO_2_ with different TiO_2_ shell thicknesses are set as 55 nm, 60 nm, 65 nm, 70 nm and 80 nm; The dipole ratios are 11:10, 6:5, 13:10, 7:5, 8:5; The refractive index is set to 1.

## 3. Results and Discussions

The phase composition and crystallographic structure of the samples were confirmed by XRD measurement and results are given in Figure 1. The pure CWO presented hexagonal structure of Cs_0.32_WO_3_ (JCPDS 83-1334), and no impurity peaks were observed in the pattern. For TiO_2_-coated CWO nanoparticles, the extra peaks appeared in the pattern which belongs to TiO_2_ (JCPDS 21-1272). With the increase in the amount of Ti source, the peak intensity at 25° is obviously increasing, indicating that the content of TiO_2_ is increasing. These XRD results indicate that there are both CWO and TiO_2_ crystals in the sample. The two structures exist independently and the formation of TiO_2_ did not affect the structure of CWO.

Figure 2 shows the SEM images of uncoated CWO and TiO_2_-coated CWO nanoparticles. The element mapping and corresponding element spectrum for CWO@0.2TiO_2_ are also given in Figure 2. The uncoated CWO is composed of irregular particles with good dispersion, and the size is about tens of nanometers. Unlike pure CWO samples, it is clearly seen that a layer of several nanometer-sized small particles appeared on the surface of the CWO particles for the TiO_2_ coated samples. With the increase in TiO_2_ content, CWO@0.6TiO_2_ exhibits obvious spherical shape with largest size. Combined with the XRD results, it can be concluded that CWO is coated with a layer of TiO_2_. The element mapping of all samples and corresponding element spectrum for CWO@0.2TiO_2_ are also given in Figure 3, which illustrate that Cs, W, O and Ti elements are uniformly distributed in the selected area on coated samples, and no other elements are found.

Figure 4 shows the TEM images of coated CWO samples. It can be clearly seen in Figure 4a–c that tens of nanometers of particles are coated by several nanometers of small particles, and the small particles in the outer layer are increasing with the increase in Ti source. Figure 4d clearly exhibits the single-crystalline nature of the two kinds of particles. The lattice fringes of d = 0.38 nm have good agreement with the (002) crystal planes of CWO structure shown in Figure 1, while the lattice fringe of d = 0.35 nm corresponds with the (101) crystal plane of tetragonal TiO_2_ phase. In Figure 4e,f, the diffraction rings of TiO_2_ such as (101), (004) and (220) are obtained, which is consistent with the XRD analysis (JCPDS 21-1272). The TEM results confirm that the surface of CWO is coated with a layer of crystalline TiO_2_.

The chemical states of the coated samples was carefully determined by XPS. The full range XPS spectra and W_4f_ core-level spectra of CWO@0.2TiO_2_ are given in Figure 5. Except the existence of Cs, W, O, and Ti elements, no other impurity elements were found in the full range XPS spectra, which is consistent with the element spectrum results in Figure 3e. The W_4f_ core-level XPS spectra of CWO@0.2TiO_2_ can be fitted as two spin-orbit doublets. The peaks at 37.3 and 35.2 eV were attributed to W^6+^, and the peaks at 36.1 and 33.8 eV were attributed to W^5+^, respectively. The NIR shielding properties of CWO are determined by the plasmon resonance of free electrons. The addition of Cs element into WO_3_ structure can reduce part of W^6+^ to W^5+^, so as to enhance the carrier concentration and small polaron mechanism [22], which is the reason why CWO material has high NIR shielding performance.

In order to observe the stability of coated CWO in the actual environment, the obtained powders were dispersed in ethanol and made into thin films on glass slides to test their absorption behavior. Figure 6a shows the absorption curve of CWO and CWO@0.2TiO_2_ nanoparticles after different durations. For the uncoated CWO nanoparticles, the NIR absorption ability degraded seriously after 30 days. While the NIR absorption of CWO@0.2TiO_2_ nanoparticles showed only slight degradation, indicating that TiO_2_ showed a good protective effect that TiO_2_ prevents Cs^+^ escape from the particle surface and form WO_3_. To determine the effect of different Ti sources content on the transmittance behavior of CWO particles, the three CWO@TiO_2_ samples were dispersed with PVB resin and coated on a glass slide (size of 10 cm × 10 cm) using a spin coating method. By observing the CWO@0.2TiO_2_ coated glass photo in Figure 6b, no opaque or haze-like property was found. The transmittance curve of three CWO@TiO_2_-coated glass are revealed in Figure 6b, and the transmittance curve of CWO-coated glass is also given for comparison purpose. It can be clearly seen that the four samples show good transparent NIR shielding properties. The transmittance in the visible light decreases with the increase in Ti source and increases obviously in the NIR region. This shows that the thicker the TiO_2_ shell, the worse the transparent heat shielding performance. This can be attributed to two reasons, one is the influence of the intrinsic optical properties of TiO_2_, and the other is the increasing particle size with the increase in TiO_2_ shell thickness. Our previous research shows that the larger the particle size of transparent heat shielding material, the worse the NIR shielding performance [15].

However, it is difficult to accurately control the particle size, shell thickness and other factors in the experiment. In order to systematically study the effect of shell thickness on the optical properties of CWO, we calculated the extinction characteristics of CWO with different TiO_2_ thickness by using the discrete dipole approximation (DDA) method. Figure 7 presents the extinction efficiencies of CWO spherical particles with different TiO_2_ shell thicknesses. It is discernible that uncoated CWO shows high extinction in the NIR region and low extinction in the visible region, indicating the transparent heat-shielding properties of CWO materials. With the increase in TiO_2_ thickness, the absorption edge in the visible region is red-shifted, and the extinction in the NIR region is weakened. We infer that the red shift of the absorption edge in the visible region is related to the intrinsic optical properties of TiO_2_, while the weakening of the extinction in the NIR region is related to the increase in particle size. Such a trend of optical response with TiO_2_ shell thickness obtained by DDA calculation is in good agreement with the experimental results in Figure 6b, indicating that the shell thickness should not be too thick when coating CWO with TiO_2_.

Finally, to verify the temperature control effect of CWO@TiO_2_-coated glass, a model house has been designed to test the temperature change as shown in the Figure 8a. The CWO@0.2TiO_2_-coated glass is placed on the center of a cement wall and directly irradiated by the light from a NIR lamp (PHILIPS, R125) 50 cm away. Two thermocouples of T1 and T2 are placed behind the glass where they are directly illuminated by light and behind the cement wall where not directly illuminated by light, respectively. In addition, a blank glass was used in the control group test. Figure 8b shows the temperature changes with time measured by T1 and T2 in the model house. The results show that the CWO@0.2TiO_2_ coated glass reduces the T1 and T2 temperature by 6.3 °C and 2.5 °C, respectively, indicating that TiO_2_-coated CWO still has good heat shielding effect. However, although the other two samples CWO@0.4TiO_2_ and CWO@0.6TiO_2_ also have the cooling effect, the effect is not as good as CWO@0.2TiO_2_, because the thicker the coating, the greater the near-infrared transmittance, which is consistent with the results in Figure 6b.

## 4. Conclusions

In this article, nanocrystalline CWO particles were prepared by solvothermal method and coated with small TiO_2_ crystals. The XRD, SEM and TEM results show that the surface of CWO is coated with a layer of crystalline TiO_2_, and the thickness of TiO_2_ shell increases with the increase in TTIP amount in the reaction process. The absorption spectrum illustrates that the NIR absorption stability of CWO@0.2TiO_2_ is much better than that of CWO after 30 days, indicating that TiO_2_ coating significantly improves the stability of tungsten bronze. The transmittance of CWO@TiO_2_-coated glass in the visible region decreases with the increase in Ti source and increases obviously in the NIR region, indicating that the thicker the TiO_2_ shell, the worse the transparent heat shielding performance. The DDA simulation results also confirm this trend. The measurement of temperature control effect in the model house gives that the CWO@0.2TiO_2_-coated glass reduces the indoor temperature by 6.3 °C and 2.5 °C at different places in the room, respectively, which demonstrated the good heat shielding performance of TiO_2_-coated CWO.

## Figures and Tables

**Figure 1 nanomaterials-12-02806-f001:**
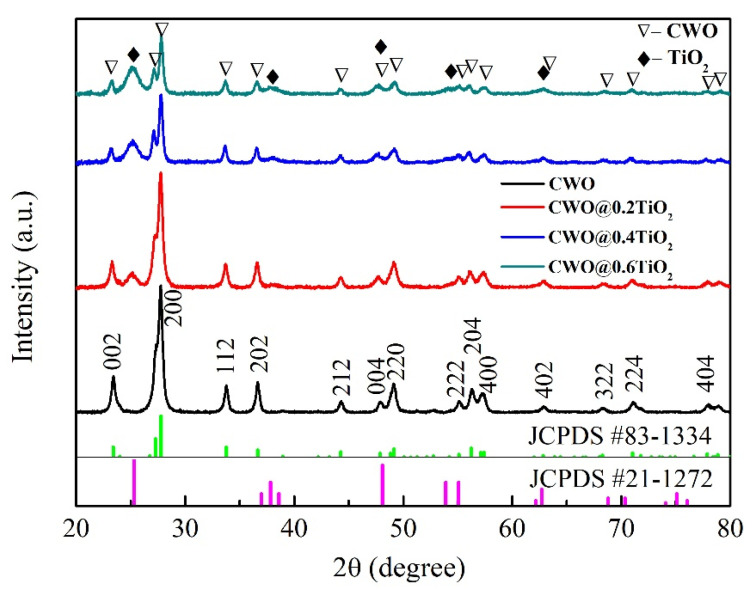
XRD patterns of the uncoated Cs_x_WO_3_ and TiO_2_ coated Cs_x_WO_3_ nanoparticles.

**Figure 2 nanomaterials-12-02806-f002:**
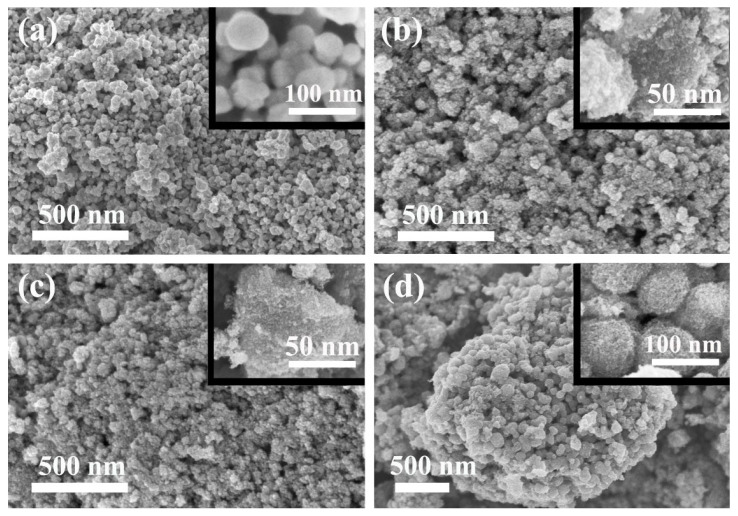
SEM image of the (**a**) CWO, (**b**) CWO@0.2TiO_2_, (**c**) CWO@0.4TiO_2_, (**d**) CWO@0.6TiO_2_ (inset shows a magnification of one segment).

**Figure 3 nanomaterials-12-02806-f003:**
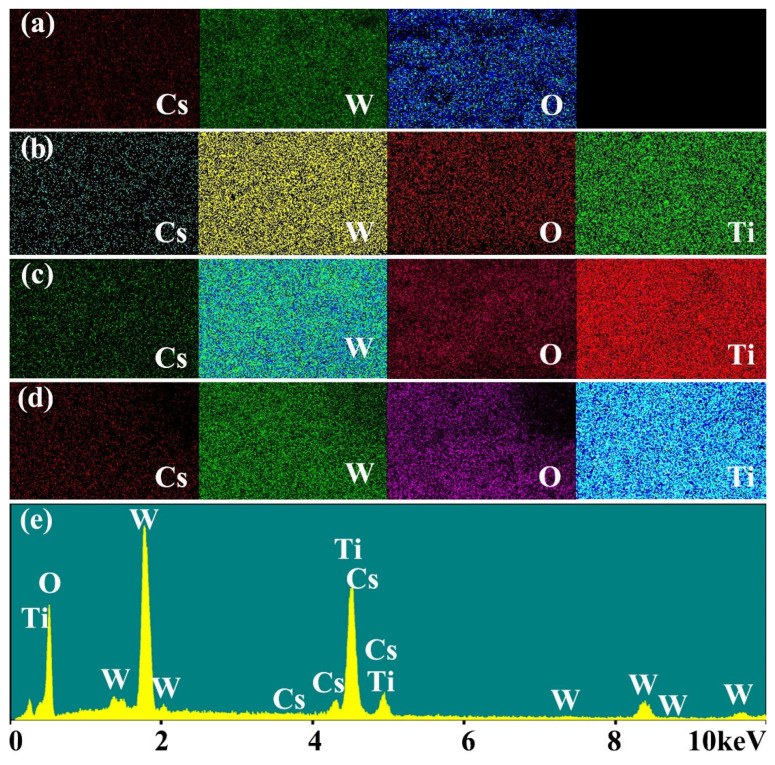
Element mapping of (**a**) CWO, (**b**) CWO@0.2TiO_2_, (**c**) CWO@0.4TiO_2_, (**d**) CWO@0.6TiO_2_; (**e**) Element spectrum of CWO@0.2TiO_2_.

**Figure 4 nanomaterials-12-02806-f004:**
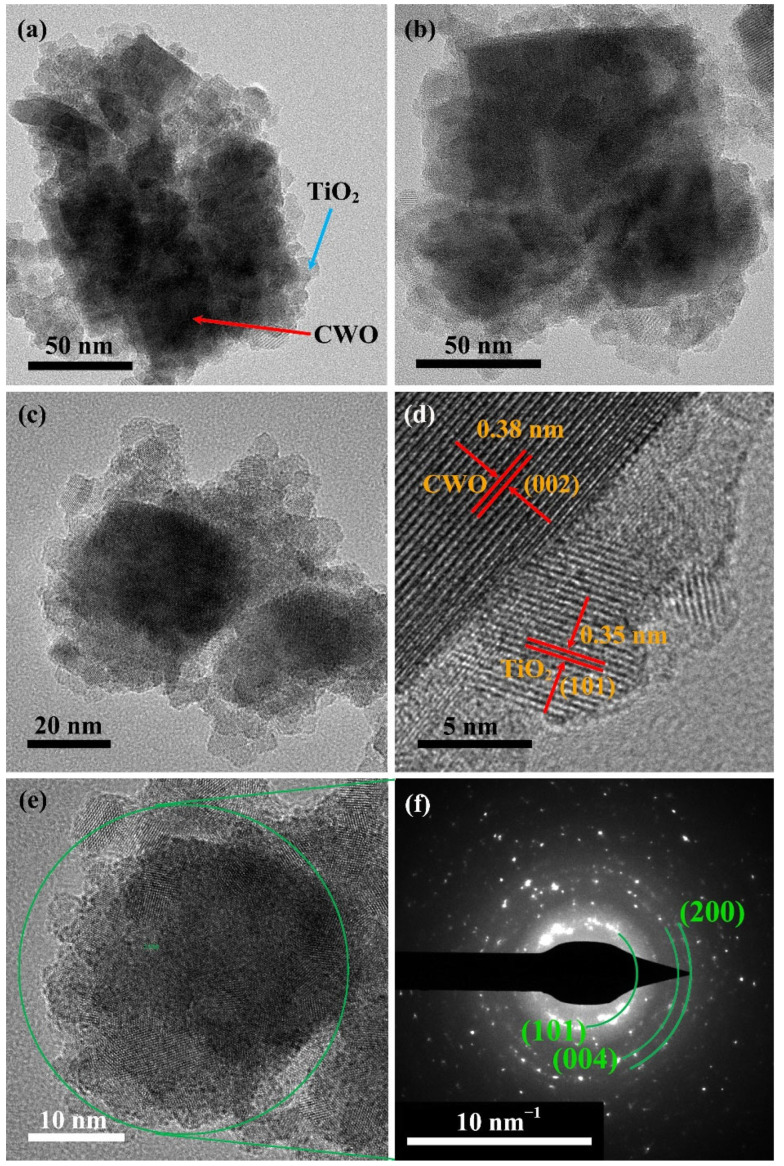
TEM images of the (**a**) CWO@0.2TiO_2_, (**b**) CWO@0.4TiO_2_, (**c**) CWO@0.6TiO_2_; HRTEM image of the (**d**) CWO@0.2TiO_2_; SAED of the CWO@0.2TiO_2_ (e) selected area, (f) diffraction rings.

**Figure 5 nanomaterials-12-02806-f005:**
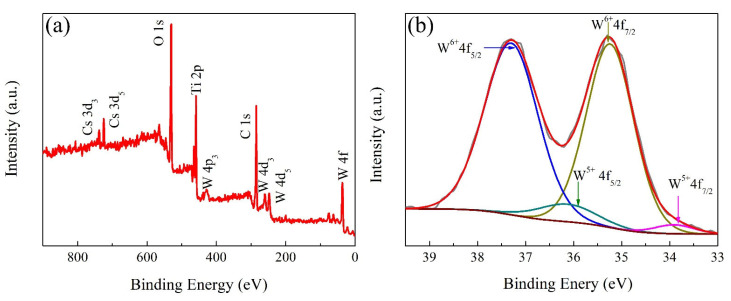
(**a**) Full range XPS spectra and (**b**) W_4f_ core-level XPS spectra of CWO@0.2TiO_2_.

**Figure 6 nanomaterials-12-02806-f006:**
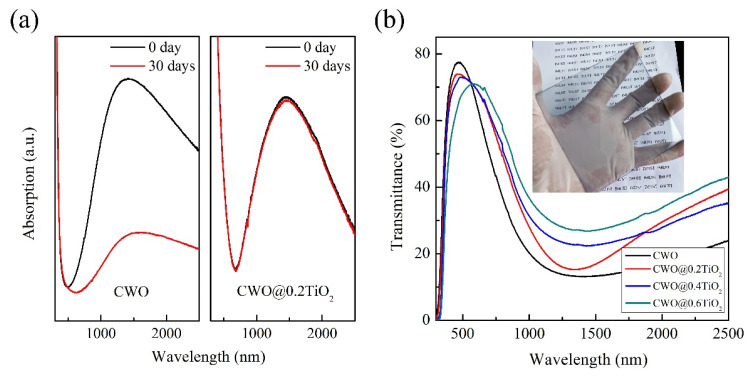
(**a**) The absorption curve of CWO and CWO@0.2TiO_2_ nanoparticles after different durations, (**b**) Transmittance curve of coated glass (inset shows the CWO@0.2TiO_2_-coated glass used in the transmittance test).

**Figure 7 nanomaterials-12-02806-f007:**
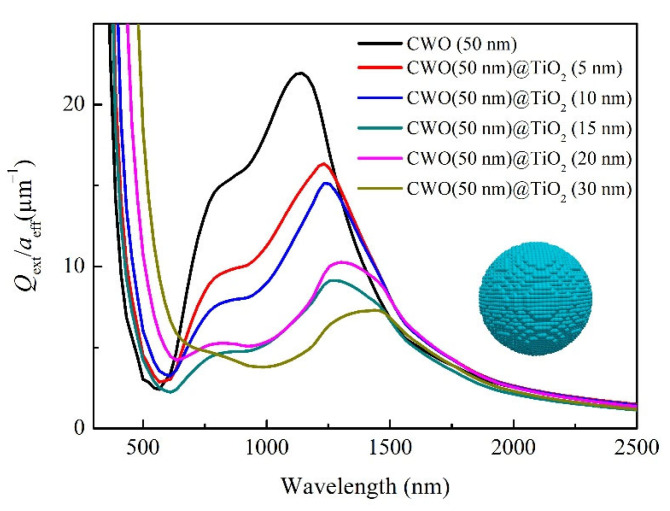
Extinction efficiencies of CWO spherical particles with different TiO_2_ shell thicknesses.

**Figure 8 nanomaterials-12-02806-f008:**
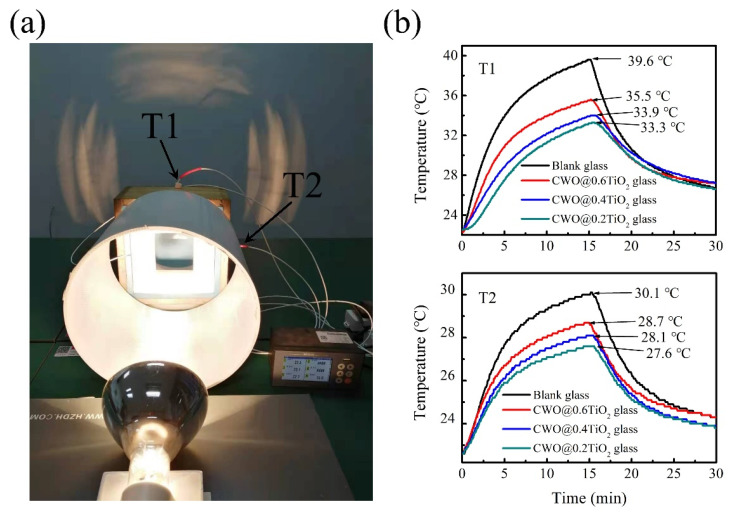
(**a**) Photographs of devices used in temperature control test, (**b**) The temperature changes with time measured by T1 and T2 in the model house.

## Data Availability

No data reported other than that presented.

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
