# Peer review of "Transparent Heat Shielding Properties of Core-Shell Structured Nanocrystalline CsxWO3@TiO2"

_nanomaterials, 2022, doi:10.3390/nano12162806_

Round 1

Reviewer 1 Report

The manuscript entitled "Transparent heat shielding properties of core-shell structured nanocrystalline CsxWO3@TiO2" reports the heat shielding properties of heterostructure nanoparticles. The authors describe the very detail fabrication and operation of the heat shielding electrodes. It is quite interesting to use the NP matrix in the heat shielding devices. However, the conclusion is not quite convincing. These results should be of interest to the inorganic and heat-shielding community, but the author should provide more evidence for their conclusion. Thus, I recommend publishing this manuscript in Nanomaterials after major revisions.

(1)    SEM images are not clear to understand the change of shell structures after the addition of the Ti precursors. Moreover, inset images are hard to understand and need bar scale information. Can the author show other SEM images and elemental mapping images of all the samples for clear verification?

(2)    Additional TEM results are required. SAED and TEM EDX mapping images would be helpful to demonstrate their core-shell structure.

(3)    In Figure 5b, why does the sample ([email protected]) show the different transmittance results (1000~2000 nm) such as FWHM values compared to the other CWO/TiO2 samples.

(4)    For Figure 7, for the conclusion, the author should provide the temperature change results with other CWO/TiO2 samples and the heat-shielding properties after different durations. This comparison can verify the need of the core-shell structures.

Reviewer 2 Report

The authors show the synthesis of Cs:WO3@TiO2 nanoparticles and study their morphological, structural and optical properties. Most importantly, the stability of the Cs:WO3 nanoparticles has been improved in ambient environments. Finally, the heat shielding capability of these materials has been studied.

The work is nice and sound. I just have one remark: for the DDA calculations, not sufficient information on the input parameters has been given. Please, include in the methods section.

Round 2

Reviewer 1 Report

The authors properly revised the aforementioned request in the revised manuscript. The article can be accepted with the revised version.